# INSTRUCTENGINE: INSTRUCTION-DRIVEN TEXT-TO-IMAGE ALIGNMENT

## ABSTRACT

Reinforcement Learning from Human/AI Feedback (RLHF/RLAIF) has been extensively utilized for preference alignment of text-to-image models. Existing methods face certain limitations in terms of both data and algorithm. For training data, most approaches rely on manual annotated preference data, either by directly fine-tuning the generators or by training reward models to provide training signals. However, the high annotation cost makes them difficult to scale up, the reward model consumes extra computation and cannot guarantee accuracy. From an algorithmic perspective, most methods neglect the value of text and only take the image feedback as a comparative signal, which is inefficient and sparse. To alleviate these drawbacks, we propose the **InstructEngine** framework. Regarding annotation cost, we first construct a taxonomy for text-to-image generation, then develop an automated data construction pipeline based on it. Leveraging advanced large multimodal models and human-defined rules, we generate 25K text-image preference pairs. Finally, we introduce cross-validation alignment method, which refines data efficiency by organizing semantically analogous samples into mutually comparable pairs. Evaluations on DrawBench demonstrate that InstructEngine improves SD v1.5 and SDXL's performance by 10.53% and 5.30%, outperforming state-of-the-art baselines, with ablation study confirming the benefits of InstructEngine's all components. A win rate of over 50% in human reviews also proves that InstructEngine better aligns with human preferences.

## 1 INTRODUCTION

Advancements in text-to-image generation are pushing the boundaries of AIGC Croitoru et al. (2023); Yang et al. (2023). Text-to-image models, especially diffusion-based models Ho et al. (2020), can generate appealing images that align with input texts to meet user intentions, garnering substantial research and application interest. Although large-scale pre-trained text-to-image generators exhibit impressive performance Podell et al. (2024); AI (2025), challenges such as text-image consistency, aesthetics, and image distortion continue to persist Xu et al. (2023a). Inspired by the training paradigm of large language models (LLMs) Grattafiori et al. (2024); Liu et al. (2024), which typically follow a pre-train + post-train approach, several RLHF-style alignment methods Lee et al. (2024); Zhang et al. (2024a); Miao et al. (2024) have been proposed to tackle above questions for text-to-image generation.

Most of these methods rely on annotated data to align text-to-image models with user preferences. For example, Diffusion-DPO Wallace et al. (2024a) takes positive/negative image pair from online users to optimize diffusion models. Although training with manual data is effective, the cost of annotation hinders the scaling of preference data and fine-grained preference modeling. The emergence of reward models Xu et al. (2023a); Liang et al. (2024) partially addresses these issues by scoring images with models rather than annotators. Reward models can evaluate whether generated images align with different dimensions of human preferences and provide reward scores. However, the preference modeling of reward models also relies on annotated data, training and inference of reward models introduce additional costs. Besides, their reward scores lack interpretability and can cause bias Wu et al. (2023a). Another drawback of existing methods is that, whether for training generators or reward models, the construction of preference data focuses solely on the evaluation of images, neglecting the significance of the text side in preference modeling. PRIP Zhan et al. (2024)

Figure 1: Differences in preference modeling: Previous alignment methods convey preferences through preferred/disliked images or reward model, InstructEngine framework encodes fine-grained preference information in three dimensions through text, making the injected preferences understandable by humans.

and PAE Mo et al. (2024) has shown that refining input text can lead to substantial enhancements for image generation. However, they have not attempted to improve the generator.

Recognizing the potential of text, we propose to alleviate above drawbacks with a new paradigm for text-to-image alignment: Injecting multifaceted preference information through fine-grained instructions. We present InstructEngine framework, which consists of: (1) Taxonomy for image-to-text instructions. Combining LLM prompt engineering and rigorous human review, our instruction taxonomy divides text-to-image scenarios into 33 major themes, each containing 20+ subtopics. We generate instructions based on it to ensure diversity. (2) Automated preference data construction pipeline: For all subtopics, we first generate coarse-grained base instructions, then add opposite details of three dimensions to construct preference instruction pairs. Finally, we generate images consistent with these instructions. We construct 25K samples for alignment training with low requirement for human annotation. (3) Cross-validation alignment algorithm. Given samples consists of two semantically analogous insturctions and two corresponding images, we select the appropriate triples to calculate multiple DPO loss. With paired instructions serving as validation for each other, generators can learn fine-grained preference information efficiently.

We conduct automated and human evaluation on DrawBench to test the efficacy of InstructEngine: After alignment, the average performance of SD v1.5 and SDXL improves by 10.53% and by 5.30%, surpassing the suboptimal baseline by 1.47% and by 1.83%. In the human evaluation, InstructEngine beats all baselines with a win rate over 50%.

We summarize the contributions of this paper as follows:

(1) We identify several limitations of existing text-to-image alignment methods: Reliance on annotated preference data, lack of interpretability in preference modeling, and insufficient utilization of text for preference alignment.

(2) We propose InstructEngine framework to alleviate these shortcomings. InstructEngine consists of a text-to-image taxonomy for data efficiency and diversity, an automated data construction pipeline to inject fine-grained preferences through differentiated instructions, making preference modeling interpretable, and a cross-validation alignment algorithm to construct multiple preference pairs from a single sample, increasing sample efficiency.

(3) InstructEngine demonstrates excellent efficiency and efficacy: With only 25K data samples for alignment training, which is much smaller than that of other methods, the two trained models achieve superior performance compared to SOTA baselines in automated evaluation and human review. Our results confirm the significant potential of instructions for text-to-image alignment.

## 2 RELATED WORK

**Text-to-Image Generation.** Text-to-image generation has evolved significantly with the advent of deep generative models. Early approaches rely on Generative Adversarial Networks (GANs) Esser et al. (2021); Zhou et al. (2022) and auto-regressive architectures Ding et al. (2021; 2022); Yu et al. (2022). Recently, diffusion models Podell et al. (2024); Rombach et al. (2022b); Saharia et al. (2022) that transform multi-mode distributions into the standard Gaussian have dominated the field Dhariwal & Nichol (2021) due to their exceptional fidelity and diversity. Diffusion methods can divided into two categories: latent-based and pixel-based. Latent-based methods, such as Stable Diffusion Podell et al. (2024); Face (2024), leverage auto-encoders to operate in compressed latent spaces, enabling efficient high-resolution synthesis. Pixel-based approaches, including DALL·E 2 Ramesh et al. (2022) and Imagen Saharia et al. (2022), directly model the image space, often integrate large language models for enhanced semantic alignment.

While evaluation metrics such as Inception Score (IS) Salimans et al. (2016), Fréchet Inception Distance (FID) Heusel et al. (2017) and CLIP score Radford et al. (2021) provide quantitative measures of image quality and text alignment, recent efforts emphasize semantic coherence generation. Works like Attend-and-Excite Chefer et al. (2023) enhance semantic guidance in diffusion models through attention mechanisms, while Instruct-Imagen Hu et al. (2024) leverages multi-modal instructions for precise control. However, achieving robust alignment between text prompts and generated images remains challenging, particularly. Recent work has focused more on human feedback, evaluating the generated results with user satisfaction Dong et al. (2023).

**Learning from Human Feedback.** Incorporating human feedback into generative models has proven pivotal for aligning outputs with user intent, as demonstrated by Reinforcement Learning from Human Feedback (RLHF) in large language models Ouyang et al. (2022); Bai et al. (2022). For text-to-image generation, RLHF-inspired approaches aim to bridge the gap between statistical metrics and human preferences. ImageReward Xu et al. (2023b) trains a reward model on human-annotated data to guide diffusion models toward aesthetically pleasing outputs. Similarly, PickScore Kirstain et al. (2023a) leverages large-scale user preferences collected via interactive platforms, while HPS Wu et al. (2023b) and its successor HPS v2 Wu et al. (2023a) curate datasets to align generated images with holistic human judgments.

Recent innovations extend RLHF to multi-dimensional preference learning Zhang et al. (2024a). VisionReward Xu et al. (2024) introduces a fine-grained evaluation framework that decomposes human preferences into interpretable dimensions, mitigating biases inherent in single-score metrics. Direct Preference Optimization (DPO) Rafailov et al. (2023b) methods represented by Diffusion-DPO Wallace et al. (2024b) bypass reward modeling. However, these approaches risk over-optimizing specific attributes at the expense of others. To address this, DRaFT Clark et al. (2023) employ multi-objective reinforcement learning, balancing diverse rewards during fine-tuning and Parrot Lee et al. (2024) tries to reach pareto-optimal with multiply rewards. A key challenge lies in scaling human feedback collection while preserving diversity Zhang et al. (2025). While datasets like AGIQA-1k Zhang et al. (2023b) and AGIQA-3k Li et al. (2023) annotate both overall quality and text-image alignment, their limited size restricts generalization. Recent work Lin et al. (2024); Zhang et al. (2023a) leverages Vision-Language Models (VLMs) to simulate human judgments, though their accuracy remains suboptimal compared to specialized reward models. These advancements underscore the importance of integrating multi-faceted human feedback into training pipelines to achieve robust, user-aligned text-to-image generation.

## 3 INSTRUCTENGINE

### 3.1 TEXT-TO-IMAGE TAXONOMY

In this section, we introduce the InstructEngine taxonomy. Our taxonomy comprehensively covers text-to-image scenarios, serving as the basis for diverse prompt construction.

**Taxonomy Construction.** Inspired by TaskGalaxy Chen et al. (2025), which develops diverse multi-modal understanding tasks, demonstrating high efficiency and significant improvements, we combine LLM agents and human labor for taxonomy construction. InstructEngine taxonomy construction process consists of three steps: (1) Seed theme set construction: We randomly sample 1,000

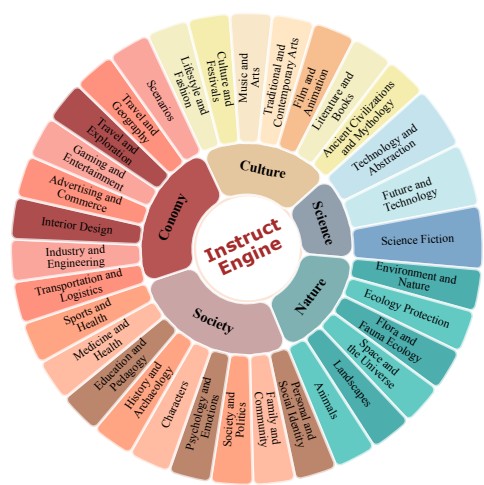

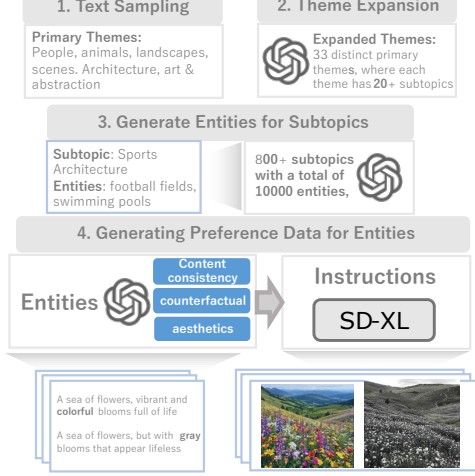

Figure 2: Visualization of themes in In-structEngine's taxonomy.

Figure 3: Construction pipeline of In-structEngine's preference data.

input texts from the Pick-a-pic Kirstain et al. (2023b) dataset and manually categorize them into six primary themes: *people*, *animals*, *landscapes*, *scenes*, *architecture*, and *art & abstraction*. (2) Primary theme expansion: We design instructions to prompt GPT-4o to expand the seed theme set. After the expansion, we obtain 33 distinct primary themes. (3) Subtopic division: We further prompt GPT-4o to divide each primary theme into 20+ subtopics based on dimensions such as themes, styles, purposes, time and space, and vertical domains. For example, *animals* are divided based on space into *savanna*, *forest*, and *domestic* animals. They are also categorized based on actions into *running*, *flying*, *swimming*, and *feeding* animals.

**Quality Control.** During theme expansion and subtopic division, GPT-4o initially generates multi-ple responses to ensure diversity. The rationality and independence of each theme and subtopic are then manually reviewed. For semantically repetitive themes and subtopics, annotators merge them to ensure the taxonomy's conciseness and accuracy.

### 3.2 PREFERENCE DATA CONSTRUCTION PIPELINE

To inject multifaceted preference information through instructions, we first construct various fine-grained preference instructions based on all subtopics in our taxonomy and then generate the corre-sponding images as preference data.

**Coarse-grained Instructions Construction.** For each subtopic in the taxonomy, we first instruct GPT-4o to provide multiple specific entities belonging to that subtopic. Then, we use CLIP embed-ding to filter out entities with low text similarity to the subtopic. Finally, GPT-4o confirms one by one whether the retained entities belong to the subtopic. In this way, we equip 800 subtopics with a total of 15,000 entities. For example, the subtopic *Sports Architecture* contains entities including *football fields*, *swimming pools*, and *climbing gyms*. These entities serve as coarse-grained base instructions, defining the content for each sample.

**Fine-grained Preference Injection.** We add multifaceted descriptions to the base instructions to generate fine-grained preference instructions. For each base instruction, we provide contrasting descriptions from three dimensions: content consistency, counterfactual scenarios, and aesthetics, forming paired preference instructions. The content consistency divergences refer to the variations in object type, quantity, physical attributes, etc., between two instructions, like *"A model showcases new accessories, wearing a necklace and a pair of earrings"* and *"Multiple models showcase new fashion, none of them wears accessories"*. The counterfactual divergences refer to the inclusion of imaginary or distorted content in one of the instructions, like *"An indoor swimming facility with a standard swimming pool and regular chairs"* and *"An indoor swimming facility with a pool smaller than a bathtub and gigantic chairs"*. Aesthetic divergence refers to two instructions having dif-ferent aesthetic styles, like *"A sea of flowers, vibrant and colorful blooms full of life"* and *"A sea of flowers, but with gray blooms that appear lifeless"*. These three kinds of divergences respec-

tively inject preferences in terms of image-text consistency, authenticity, and aesthetics. In this way, the preferences are represented by the differences between each pair of instructions, making them interpretable. We ultimately obtained 26,430 pairs of preference instructions with 15,000 entities.

**Preference Data Generation.** After constructing the preference instructions, we take a foundation text-to-image model, SDXL Podell et al. (2024), to generate matching images for each fine-grained instruction. Since we primarily focus on exploring preference information injection through text, we pay more attention to ensure the consistency between the generated images and the text than the image quality. We first apply SDXL to generate 8 images for each instruction with different random seeds. Then we take BLIP Li et al. (2022) to pick the image that best matches the instruction from 8 images. At last, GPT-4o serves as a judge to filter out mismatched image-text samples. Ultimately, we pick out 24,716 data instances from the 26,430 instruction pairs. Each instance contains two contrasting instructions with fine-grained differences and two corresponding images.

### 3.3 CROSS-VALIDATION ALIGNMENT TRAINING

#### 3.3.1 BACKGROUND OF DPO AND DIFFUSION-DPO

The main idea of Direct Preference Optimization (DPO) Rafailov et al. (2023a) is to integrate the reward modeling loss into the LLM's training loss, thereby eliminating the separate reward model to simplify the alignment process. Specifically, given condition $\mathbf{c}$, winning/losing output $\mathbf{y}_w/\mathbf{y}_l$, the Bradley-Terry loss for the reward function $r(\mathbf{c}, \mathbf{y})$ with parameter $\phi$ is:

$$L_{\text{BT}}(\phi) = -\mathbb{E}_{\mathbf{c},\mathbf{y}^w,\mathbf{y}^l}\left[\log \sigma\left(r_\phi(\mathbf{c}, \mathbf{y}^w) - r_\phi(\mathbf{c}, \mathbf{y}^l)\right)\right], \tag{1}$$

The RLHF loss for LLM parameter $\theta$ under dataset $\mathcal{D}_c$ is:

$$\max_{p_\theta} \mathbb{E}\left[r(\mathbf{c}, \mathbf{y})\right] - \beta\mathbb{D}_{\text{KL}}\left[p_\theta(\mathbf{y}|\mathbf{c})\|p_{\text{ref}}(\mathbf{y}|\mathbf{c})\right], \tag{2}$$

where $\mathbf{c}\sim\mathcal{D}_c$, $\mathbf{y}\sim p_\theta(\mathbf{y}|\mathbf{c})$ (conditional distribution), $\beta$ controls KL-divergence regularization from reference model $ref$. With the unique global optimal solution $p_\theta^*$ and partition function $Z(\mathbf{c}) = \sum_{\mathbf{y}} p_{\text{ref}}(\mathbf{y}|\mathbf{c})\exp\left(r(\mathbf{c}, \mathbf{y})/\beta\right)$, the reward function $r(\mathbf{c}, \mathbf{y})$ can be rewritten as:

$$r(\mathbf{c}, \mathbf{y}) = \beta\log\frac{p_\theta^*(\mathbf{y}|\mathbf{c})}{p_{\text{ref}}(\mathbf{y}|\mathbf{c})} + \beta\log Z(\mathbf{c}). \tag{3}$$

Replace $r_\phi(\mathbf{c}, \mathbf{y})$ in Eq. (1) with Eq. (3) and we get:

$$L_{\text{DPO}}(\theta) = -\mathbb{E}_{\mathbf{c},\mathbf{y}^w,\mathbf{y}^l}\left[\log \sigma\beta\left(\log\frac{p_\theta(\mathbf{y}^w|\mathbf{c})}{p_{\text{ref}}(\mathbf{y}^w|\mathbf{c})} - \log\frac{p_\theta(\mathbf{y}^l|\mathbf{c})}{p_{\text{ref}}(\mathbf{y}^l|\mathbf{c})}\right)\right]. \tag{4}$$

For diffusion models Ho et al. (2020), given $\mathbf{y}_0\sim q(\mathbf{y}_0)$ as data distribution, the $T$-step forward process $q(\mathbf{y}_{1:T}|\mathbf{y}_0)$ adds noise $\epsilon$ to the data $\mathbf{y}_0$, reverse process $p_\theta(\mathbf{y}_{0:T})$ transits to recover $\mathbf{y}_0$. The squared difference for noise prediction is defined as:

$$\delta_\theta(\epsilon, \mathbf{y}_t, t) = \|\epsilon - \epsilon_\theta(\mathbf{y}_t, t)\|_2^2 \tag{5}$$

The training loss is to minimize the evidence lower bound:

$$L_{\text{DM}} = \mathbb{E}_{\epsilon\sim\mathcal{N}(0,\mathbf{I}),\mathbf{y}_0,t}\left[\delta_\theta(\epsilon, \mathbf{y}_t, t)\right], \tag{6}$$

$\mathbf{y}_t\sim q(\mathbf{y}_t|\mathbf{y}_0)$, timestep $t\sim\mathcal{U}(0, T)$. Give $\mathcal{D} = (\mathbf{c}, \mathbf{y}_0^w, \mathbf{y}_0^l)$, Diffusion-DPO Wallace et al. (2024a) adapts DPO to align diffusion models:

$$\begin{aligned}\Delta_w &= \delta_\theta(\epsilon_w, \mathbf{y}_t^w, t) - \delta_{\text{ref}}(\epsilon_w, \mathbf{y}_t^w, t),\\ \Delta_l &= \delta_\theta(\epsilon_l, \mathbf{y}_t^l, t) - \delta_{\text{ref}}(\epsilon_l, \mathbf{y}_t^l, t),\\ L(\theta) &= -\mathbb{E}_{(\mathbf{y}_0^w,\mathbf{y}_0^l)\sim\mathcal{D},t}\log \sigma\left[-\beta T\omega(\lambda_t)(\Delta_w - \Delta_l)\right],\end{aligned} \tag{7}$$

$\mathbf{y}_t^* = \alpha_t\mathbf{y}_0^* + \sigma_t\epsilon^*$, $\epsilon^*\sim\mathcal{N}(0, \mathbf{I})$ is a draw from $q(\mathbf{y}_t^* \mid \mathbf{y}_0^*)$. $\lambda_t = \alpha_t^2/\sigma_t^2$ is the signal-to-noise ratio, $\alpha_t$ and $\sigma_t$ are noise scheduling functions Rombach et al. (2022a), we omit $\mathbf{c}$ for compactness.

### 3.3.2 CROSS-VALIDATION PREFERENCE ALIGNMENT

**Alignment Method.** As previously introduced, each sample of InstructEngine contains two contrasting instruction-image pairs $(x_1, y_1)$ and $(x_2, y_2)$. To utilize these preference data for alignment, a natural approach is to follow the IOPO Zhang et al. (2024b) method by combining the two instructions and two images into four DPO-style preference triples: $(x_1, y_1, y_2)$, $(x_2, y_2, y_1)$, $(y_1, x_1, x_2)$ and $(y_2, x_2, x_1)$. However, although IOPO has demonstrated its superiority over DPO in pure text scenarios, it is not reasonable to directly adapt IOPO to the text-to-image task. We first introduce our cross-validation alignment algorithm and then explain why we retain only two preference triples, $(x_1, y_1, y_2)$ and $(x_2, y_2, y_1)$, that contain one instruction and two images for training.

Following Eq. (7), we calculate two pairs of $\Delta_w$ and $\Delta_l$:

$$\Delta_{\mathbf{x}_1^w} = \delta_\theta(\epsilon_{w_1}, \mathbf{y}_1, t) - \delta_{\text{ref}}(\epsilon_{w_1}, \mathbf{y}_1, t),$$
$$\Delta_{\mathbf{x}_1^l} = \delta_\theta(\epsilon_{l_1}, \mathbf{y}_2, t) - \delta_{\text{ref}}(\epsilon_{l_1}, \mathbf{y}_2, t),$$
$$\Delta_{\mathbf{x}_2^w} = \delta_\theta(\epsilon_{w_2}, \mathbf{y}_2, t) - \delta_{\text{ref}}(\epsilon_{w_2}, \mathbf{y}_2, t),$$
$$\Delta_{\mathbf{x}_2^l} = \delta_\theta(\epsilon_{l_2}, \mathbf{y}_1, t) - \delta_{\text{ref}}(\epsilon_{l_2}, \mathbf{y}_1, t),$$

In each triples, the image that aligns/dis-aligns with the instruction is designated as the winning/losing output. The two instruction-image pairs thus serve as a cross-validation for each other. And the optimization loss is defined as:

$$L(\theta) = -\mathbb{E}_{(\mathbf{x}_1, \mathbf{x}_2, \mathbf{y}_1, \mathbf{y}_2) \sim \mathcal{D}, t \sim \mathcal{U}(0, T)} \log \sigma \left[ -\beta T \omega(\lambda_t) \right.$$
$$\left. \left( \left( \Delta_{x_1^w} - \Delta_{x_1^l} \right) + \left( \Delta_{x_2^w} - \Delta_{x_1^l} \right) \right) \right], \tag{8}$$

**Explanation for Triple Selection.** We discard the other two triples for two reasons: (1) The core of text-to-image generator is to learn the mapping from text to image. However, for triples that contain two instructions and one image, the causal relationship is reversed, i.e., distinguishing the differences between texts based on the image. It is not guaranteed that this reverse causal modeling is beneficial for text-to-image generation. (2) Two contrasting instructions are generated from the same base instruction and contain some identical words. After tokenizing and encoding, the embeddings of these parts are very similar, leading to almost identical text embeddings, which can interfere with the generator's learning. In contrast, for images, although the base entity in the two images is the same, there are usually significant differences between their pixels: The randomness of the diffusion model and the different details can reduce the redundancy between the two images.

In the Appendix and Section 4.5, we provide the loss curves and evaluation metrics of using unchosen triples for training. Rapidly converging loss demonstrates that two discarded triples do not provide information and performance degradation shows their detrimental effect for training. We leave the exploration of underlying causes for future work.

## 4 EXPERIMENTS

### 4.1 SETTING

**Baselines & Datasets.** We select the following state-of-the-art alignment methods as baselines for comparison: (1) **ReFL**: ReFL adopts the reward score from the ImageReward model as the human preference loss for a latter step in the backward denoise process. The pre-training loss is retained to re-weight and regularize the preference loss. For training, we sampled 50,000 text-image pairs from DiffusionDB. (2) **Diffusion-DPO**: Diffusion-DPO takes the filtered Pick-a-pic v2 dataset for training, which contains 851,293 pairs of winning and losing images, with 58,960 unique prompts. Its training loss function is introduced in Eq. (7). (3) **HPSv2**: HPSv2 is one of the SOTA reward models for image evaluation. We separately take 58,960 prompts from Pick-a-pic v2 dataset and 49432 prompts from InstructEngine as the text source. After generating 8 images with SDXL, we choose the image with the highest HPSv2 score and the image with the lowest HPSv2 score as a pair and train the generator with the Diffusion-DPO loss.

**Training Settings.** We select Stable Diffusion v1.5 Face (2024) and SDXL Podell et al. (2024) as foundation models to compare the effectiveness of different alignment methods. For InstructEngine,

| Method | Human Preference | | | Content Consistency | | Image Quality | Average |
|---|---|---|---|---|---|---|---|
| | PickScore | ImageReward | HPSv2 | CLIP | BLIP | Aesthetic | |
| *Stable Diffusion v1.5* | | | | | | | |
| Origin | 0.2137 | -0.0170 | 0.2768 | 0.2705 | 0.4875 | 5.2369 | 1.0781 |
| ReFL | 0.2144 | 0.4262 | 0.2819 | 0.2749 | 0.4951 | 5.2557 | 1.1580 |
| Diffusion-DPO | **0.2183** | 0.3794 | 0.2820 | 0.2730 | 0.4899 | 5.3064 | 1.1582 |
| HPSv2$_{pick-a-pic}$ | 0.2160 | 0.4061 | 0.2845 | 0.2733 | 0.4926 | 5.3173 | 1.1650 |
| HPSv2$_{InstructEngine}$ | 0.2150 | 0.4563 | **0.2857** | 0.2751 | 0.4942 | 5.3285 | 1.1758 |
| InstructEngine | 0.2171 | **0.5240** | 0.2824 | **0.2870** | **0.5033** | **5.3356** | **1.1916** |
| *SDXL* | | | | | | | |
| Origin | 0.2256 | 0.6102 | 0.2863 | 0.2780 | 0.5066 | 5.5015 | 1.2347 |
| ReFL | 0.2264 | 0.7406 | 0.2902 | 0.2853 | 0.5145 | 5.5155 | 1.2621 |
| Diffusion-DPO | **0.2300** | 0.8473 | 0.2914 | 0.2715 | 0.5015 | 5.4268 | 1.2614 |
| HPSv2$_{pick-a-pic}$ | 0.2237 | 0.8128 | **0.2968** | 0.2742 | 0.5060 | 5.5118 | 1.2709 |
| HPSv2$_{InstructEngine}$ | 0.2251 | 0.7843 | 0.2957 | 0.2753 | 0.5062 | 5.5785 | 1.2775 |
| InstructEngine | 0.2285 | **0.8720** | 0.2923 | **0.2918** | **0.5259** | **5.5902** | **1.3001** |

Table 1: Performance of various text-to-image alignment methods on the DrawBench. We quantify the performance of these methods with six common metrics for images evaluation. Optimal and sub-optimal performance is denoted in bold and underlined fonts, respectively. Our method dose not achieve the highest results on the PickScore and HPSv2 because Diffusion-DPO is trained with preference data sourced from the same origin as PickScore, and HPSv2 method selects data based on the HPSv2 score.

we set the learning rate at 1e-5, the batch size at 128. Similarly to other studies, we use a constant learning rate scheduler and set the warm-up step to 0. For all baselines, we follow their reported settings, including batch size, learning rate, and other relevant settings. For methods whose hyperparameters are not provided, we set their hyperparameters the same as our InstructEngine. For fair comparison, all methods are trained for one epoch on corresponding datasets, with only the image generation module (U-Net) tunable. And all experiments are conducted in half-precision on NVIDIA A800-SXM4-80GB machines.

**Evaluation Settings.** We perform both automatic and human evaluation on DrawBench to test the performance of different methods. In the automated evaluation, we use three kinds of metrics. For *Human Preference*, we generate reward scores with reward models trained with human preference data: PickScore, ImageReward, and HPSv2. For *Content Consistency*, we apply CLIP and BLIP models to calculate the text-image matching score. For *Image Quality*, we select the Aesthetic score to reflect the aesthetic quality of the images. In the human evaluation, we require 10 annotators to compare images generated by different methods and decide which image is better or if it is a tie, considering aesthetics, safety, rationality, and consistency.

## 4.2 PRIMARY RESULTS

In Table 1, we present the performance of different alignment methods. We reach the following conclusions: (1) InstructEngine achieves optimal results in most metrics and in the overall average metric: InstructEngine improves the average metrics of SD v1.5 and SDXL on DrawBench by 10.53% and 5.30%, respectively, and outperforms the second-best method by 1.47% and 1.83%, respectively. This indicates that InstructEngine comprehensively improves the capabilities of the text-to-image generation model in terms of aesthetics, consistency, and meeting human preferences. However, InstructEngine fails to achieve optimal results in HPSv2 and PickScore. We attribute this to differences in data distribution: Diffusion-DPO trained with the Pick-a-pic v2 dataset achieves the highest PickScore, as PickScore's training data also comes from the Pick-a-pic dataset. Similarly, the two methods that use HPSv2 reward scores to filter training data achieves the highest HPSv2 scores. However, their scores are much lower on reward models from different sources. This also indicates that human preferences learned by reward models can be biased. Recognizing the risk of reward hacking, we also provide results of human evaluation to demonstrate the advantages of our method in Section 4.3. (2) Alignment training is powerful and efficient: All alignment methods can improve the foundation models' performance. After InstructEngine' alignment, the average metrics of SD v1.5 are very close to those of the original SDXL, which has a larger scale of pre-training and more parameters. This demonstrates the potential of alignment and its higher data efficiency com-

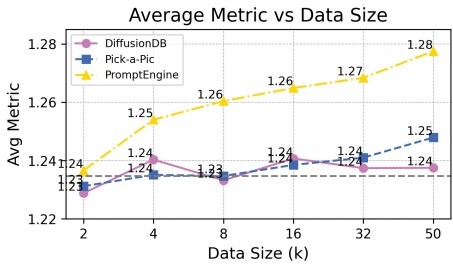

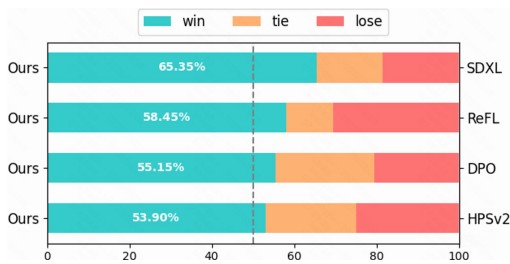

Figure 4: Data efficiency comparison across datasets. Gray dashed line: original model performance.

Figure 5: Human evaluation: InstructEngine achieves >50% win rate against baselines.

pared to pre-training. However, we also notice that the absolute improvement for SDXL is smaller compared to SD v1.5. This might be because SDXL is our data generator, and SD v1.5 effectively distills the knowledge from SDXL. (3) The instruction construction methodology of InstructEngine balances both efficiency and diversity: Among the two methods based on HPSv2, the one using InstructEngine instructions achieves higher performance. This demonstrates that our taxonomy effectively ensures coverage of text-to-image scenarios and maintains data diversity.

### 4.3 IN-DEPTH ANALYSIS

**Data Efficiency.** To verify the effectiveness of InstructEngine's instruction construction method in enhancing data efficiency, we select text prompts from Pick-a-Pic v2, DiffusionDB, and our dataset as instructions for alignment data construction. We set a data limit of 50,000 entries. Since the other two dataset do not contain paired preference instructions, our cross-validation method is not usable, so we follow the HPSv2 data construction and training process introduced in Section 4.1. We then compare the average metric values of the SDXL model across different data sources and scales. As shown in Figure 4, the model trained with our data consistently outperforms the models trained with the other two datasets across different data scales. Additionally, in the other two datasets, doubling the data scale sometimes does not lead to performance improvements and even results in declines. In contrast, the data from InstructEngine consistently yields stable gains. We attribute this to our taxonomy, which ensures data diversity and reduces redundancy. Moreover, even with only 2k data, InstructEngine still slightly improves SDXL's performance, whereas the other two datasets lead to performance degradation. The above phenomena indicate that the construction method of InstructEngine results in higher data efficiency, it also demonstrates the importance of instructions for text-to-image alignment.

**Case Study for Preference Injection.** As introduced in section 3.2, we inject three types of preference: consistency, realism, and aesthetics. Four examples in Figure 14 visually demonstrate the impact of the injected preference information on the model's generated results. In example (a), original SDXL misunderstood the prompt and generated multiple seats with wrong legs, while InstructEngine generated consistent content. In example (c), different from SDXL, InstructEngine only generated ivory in the correct parts. In examples (c) and (d), the images generated by InstructEngine are more exquisite than those generated by SDXL, especially in example (d), the generated text includes a fireworks effect. With vivid colors, fine detail, and a harmonious overall style, InstructEngine better aligns with human preferences. More comparison cases for InstructEngine and baselines are provided in the Appendix H.

### 4.4 HUMAN EVALUATION

Using SDXL as the base model, InstructEngine and four baselines generate 200 images separately in DrawBench. For HPSv2, we choose the better version with $InstructEngine$ data. Ten annotators compare the images generated by InstructEngine and those generated by other baselines in terms of consistency, aesthetics, and realism. During annotation, we obscure the image source to ensure fairness. As shown in Figure 5, InstructEngine achieves a win rate of more than 50% compared to all base models. The human evaluation result more significantly reflects the advantage of our method compared to automated metrics. We display the annotation interface in the Appendix F.

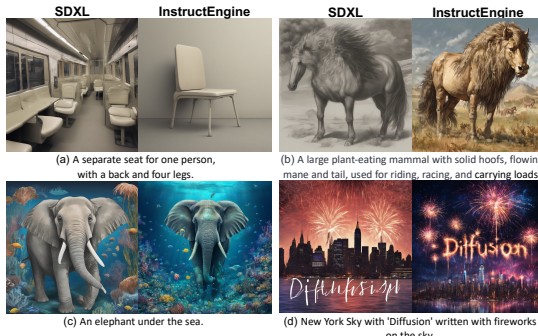

(a) A separate seat for one person, with a back and four legs.

(b) A large plant-eating mammal with solid hoofs, flowing mane and tail, used for riding, racing, and carrying loads.

(c) An elephant under the sea.

(d) New York Sky with 'Diffusion' written with fireworks on the sky.

Figure 6: The images generated by InstructEngine are significantly more aligned with the input instructions, exhibiting better realism and aesthetics.

| Method | ImageReward | HPSv2 | BLIP | Aesthetic |
|---|---|---|---|---|
| Origin | 0.6102 | 0.2863 | 0.5066 | 5.5015 |
| InstructEngine | **0.8720** | **0.2923** | **0.5259** | 5.5902 |
| w SFT | 0.6900 | 0.2923 | 0.5207 | 5.4847 |
| w/o Cross-val | 0.7277 | 0.2917 | 0.5159 | 5.4548 |
| w/o Discard | 0.1604 | 0.2816 | 0.4838 | 5.4784 |
| w Random select | 0.8220 | 0.2878 | 0.5219 | 5.5747 |
| w Flux | 0.8614 | 0.2910 | 0.5068 | **5.7191** |

Figure 7: Ablation study on each component of InstructEngine.

### 4.5 ABLATION STUDY

To validate the effectiveness of each component, we compare InstructEngine with its five variants on SDXL model: (1) Supervisely fine-tune (SFT) SDXL with loss in Eq. (6) on all text-image pairs. (2) Replace cross-validation alignment with Diffusion-DPO, instructions are not paired. (3) Retain the discarded triplets for training. (4) Randomly select images from 8 generations. (5) Take the more advanced commercial Flux-pro-1.1 model as image generator.

As shown in Figure 7, InstructEngine's each design is essential. With the same dataset, when our alignment method is replaced with SFT and DPO, the model performance decreases significantly. This indicates that our cross-validation alignment method learn preference information more efficiently. When we retain triples that contain two instructions and one image for training, there appears a significant drop in performance. We have hypothesized from two perspectives, causal modeling and modality granularity, to explain the unsuitability of such samples. We are still working on providing support for our hypotheses. Finally, we explore the impact of image quality on InstructEngine. Removing the filtering about text-image consistency causes compromised performance. This is reasonable because inconsistent images hinder the generator from learning preference information in the instructions. Due to resource constraints, we generate a single image with Flux for each instruction, this variant only shows an advantage in the Aesthetic score, since the images generated by Flux are more refined. This also indicates that our focus on text-image consistency is justified for comprehensive improvement.

## 5 CONCLUSION

We propose the InstructEngine framework to alleviate several text-to-image alignment issues including the reliance on manual annotation, the uninterpretability of reward models, and the neglect of instruction design. InstructEngine consists of an instruction taxonomy to ensure diversity, an automated data construction pipeline to reduce data annotation cost, and a cross-validation optimization algorithm to refine data efficiency. By injecting fine-grained preference information into contrasting instructions, InstructEngine performs efficient alignment: After training with 25K constructed samples, InstructEngine achieves SOTA performance. In the automated evaluation, the performance of SD v1.5 and SDXL improves by 10.53% and by 5.30%, surpassing the suboptimal baseline by 1.47% and by 1.83%. In the human review, InstructEngine achieves win rates higher than 50% over all baselines. In conclusion, InstructEngine verifies the potential of instruction for the alignment of text-to-image models. Through further experiments (Appendix G), we have verified that this method is applicable not only to diffusion-based text-to-image models but also to auto-regressive models (Emu3).

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

## A LLM Usage Declaration

In this research, Large Language Models (LLMs) were used exclusively for grammar checking and to assist with the clarity of language. No LLM was involved in the ideation or content generation processes. The authors take full responsibility for all content presented in the paper, including any generated by the LLM. We have ensured that the use of LLMs complies with ethical standards and does not constitute any form of scientific misconduct or plagiarism.

## B Ethics Statement

This work adheres to the ICLR Code of Ethics. In this study, no human subjects or animal experimentation was involved. All datasets used were sourced in compliance with relevant usage guidelines, ensuring no violation of privacy. We have taken care to avoid any biases or discriminatory outcomes in our research process. No personally identifiable information was used, and no experiments were conducted that could raise privacy or security concerns. We are committed to maintaining transparency and integrity throughout the research process.

## C Reproducibility Statement

To ensure the reproducibility of InstructEngine, we provide detailed implementation specifications throughout this work. In Section 3.1, we present the taxonomy of the data in detail. In Section 3.2, we describe the data construction process of InstructEngine comprehensively. In Section 3.3, we elaborate on the cross-validation alignment training pipeline. The experimental hyperparameters, hardware specifications and the evaluation settings are detailed in Section 4.1. Code implementation and model checkpoints will be released to facilitate the reproduction of our results.

## D SHOWCASE

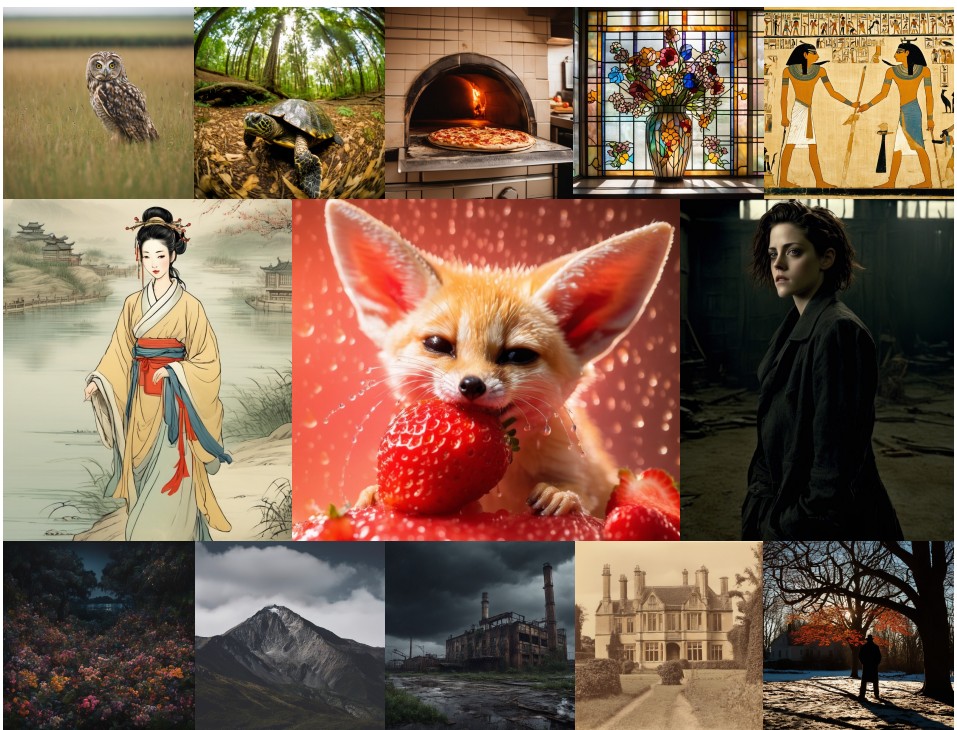

Figure 8: We propose InstructEngine, a text-to-image alignment framework injecting preference information through contrasting instructions. After training with our preference data and alignment method, the SDXL model generates images that are more realistic and align better with human aesthetic preferences. We present generation results across human, animal, artwork, food, and landscape.

## E LOSS CURVE

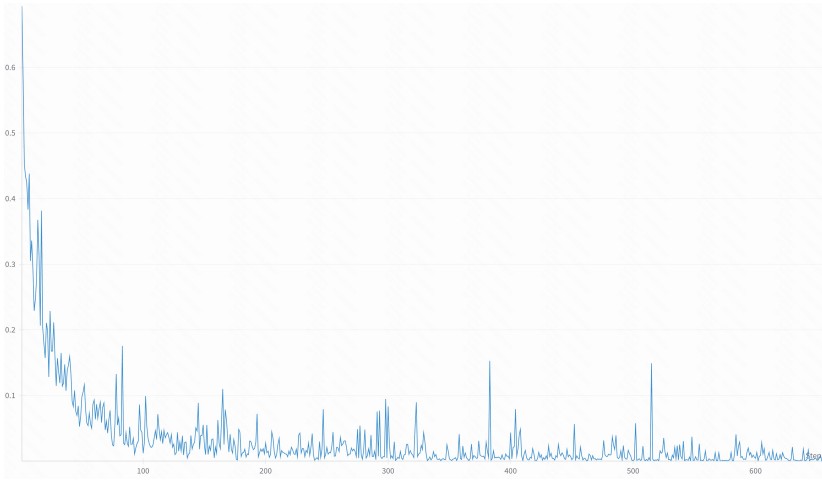

Figure 9: Loss curve for the discarded triples that contains two instructions and one image

In Figure 9 and Figure 10, We plot the resulting DPO loss curves calculated from different kinds of triples during training. In Figure 9, it can be seen that the loss from triples containing two

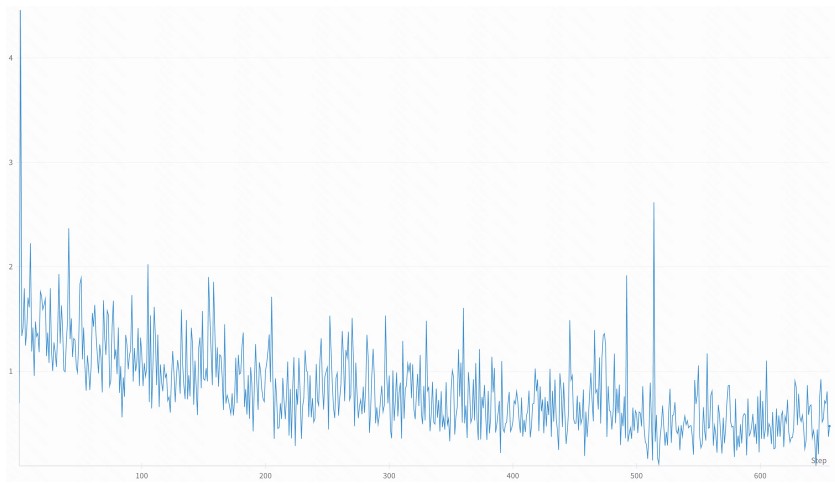

Figure 10: Loss curve for the retained triples that contains two images and one instruction

instructions as positive and negative samples for generating an image quickly converge to near 0. This indicates that they cannot form an valid contrast for the image generation model. We hypothesis that such triples' positive and negative texts can only provide preference information in text format, which can not be utilized by the generator. So we discard this kind of triples.

In Figure 10, we plot the loss curve for the retained triples that consist of one instruction and two image. During the training process, two images form a contrast, providing effective preference information for the generator to create images based on instruction information. As a result, the loss decreased slowly but did not ultimately converge to 0.

## F  DATA ANNOTATION INTERFACE.

In below Figure 11, we show the annotation interface used by annotators to compare the quality of images. To ensure fairness, image 1 and image 2 are randomly selected from our model and the baseline model.

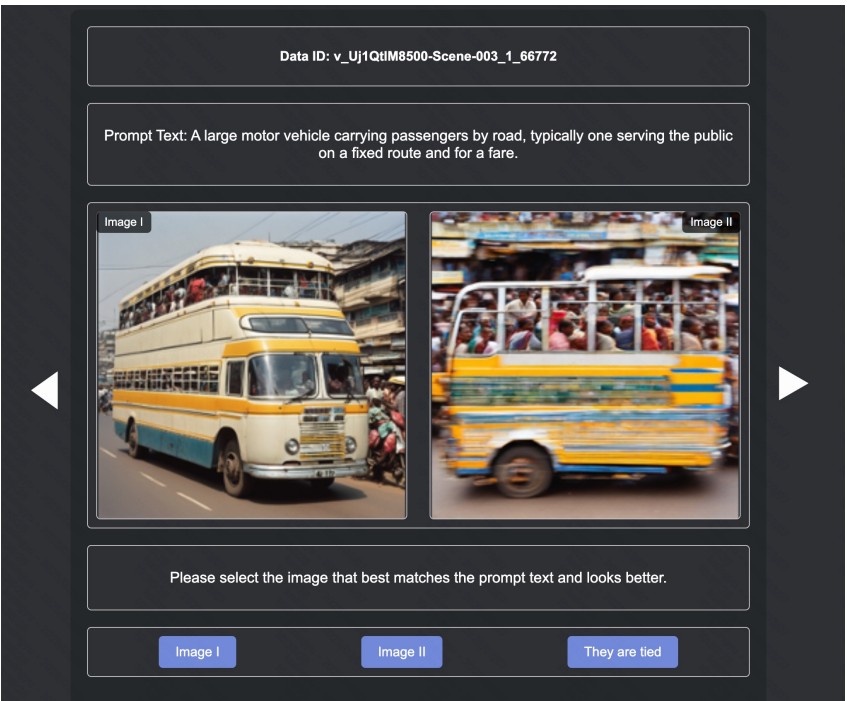

Figure 11: Annotation interface and instructions for annotators to judge which image is better aligned with the prompt text and looks better.

## G  ADAPTION

| Base | Method | T2I-CompBench++ | | | |
|---|---|---|---|---|---|
| | | Attribute | Layout | Non-spatial | Complex |
| | SUR-Adapter | 0.3426 | 0.2907 | 0.3095 | 0.3044 |
| | PAE | 0.3480 | 0.3015 | 0.2863 | 0.3239 |
| SD v1.5 | AGFSync | 0.4907 | 0.3129 | 0.3160 | 0.3314 |
| | ELLA | 0.5732 | 0.3133 | 0.3138 | **0.3436** |
| | InstructEngine | **0.6238** | **0.3176** | **0.3244** | 0.3425 |
| SDXL | AGFSync | 0.5737 | 0.3764 | 0.3181 | 0.3497 |
| | InstructEngine | **0.6315** | **0.3760** | **0.3286** | **0.3534** |
| | AGFSync | 0.5878 | 0.3614 | 0.3119 | 0.3390 |
| Emu3 | SILMM | **0.5971** | 0.3603 | 0.3051 | 0.3393 |
| | InstructEngine | 0.5883 | **0.3629** | **0.3142** | **0.3418** |

Figure 12: Evaluation Result on T2I-CompBench++

| Base | Method | DPGBench | | | | | |
|---|---|---|---|---|---|---|---|
| | | Glob | Enti | Attr | Rela | Other | All |
| | SUR-Adapter | 73.16 | 71.13 | 72.34 | 73.23 | 72.82 | 62.24 |
| | PAE | 73.24 | 72.30 | 73.59 | 73.59 | 76.00 | 62.71 |
| SD v1.5 | AGFSync | **78.14** | 76.19 | **77.33** | 79.26 | 78.05 | 67.73 |
| | ELLA | 73.46 | 73.49 | 73.20 | 77.46 | 74.24 | 63.11 |
| | InstructEngine | 77.62 | **76.27** | 75.37 | **81.55** | **80.10** | **68.55** |
| SDXL | AGFSync | 83.51 | **83.48** | 81.86 | 87.44 | 82.35 | 75.45 |
| | InstructEngine | **84.07** | 83.25 | **82.54** | **89.16** | **82.61** | **76.52** |
| | AGFSync | 83.70 | 82.42 | **84.78** | 87.79 | **69.03** | 77.39 |
| Emu3 | SILMM | 84.19 | 81.57 | 84.52 | 89.01 | 64.80 | 77.45 |
| | InstructEngine | **84.82** | **82.61** | 83.94 | **89.38** | 68.97 | **77.96** |

Figure 13: Evaluation Result on DPGBench

In Figure 12 and 13, we demonstrate that InstructEngine can be adapted to both diffusion (SD v1.5 and SDXL) and auto-regressive models (Emu3), outperforming various baselines in both T2ICompBench++ and DPGBench. This comprehensively illustrate the superiority of InstructEngine.

# H  CASE COMPARISON

In below Figure 14, we provide six prompts and the corresponding images generated by our model and the baselines. Compared to other baselines, the images generated by our model have better color contrast, text-image alignment, object realism, and aesthetic quality.

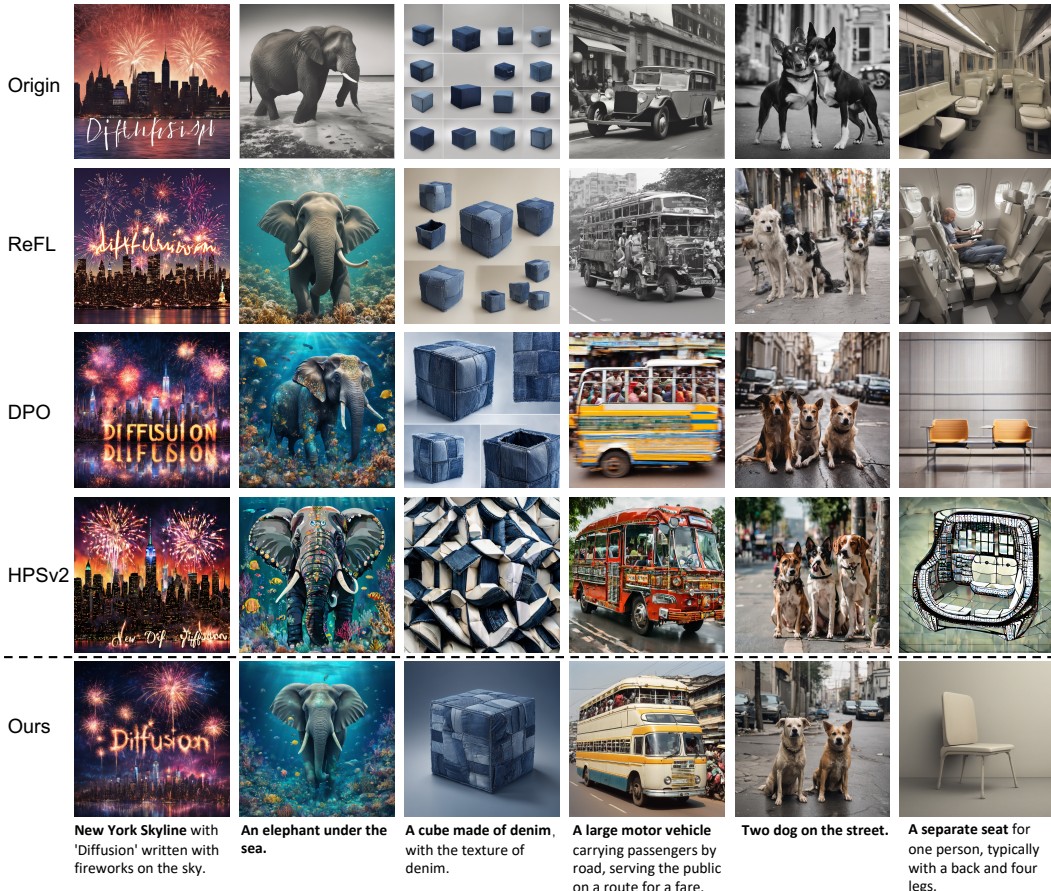

Figure 14: Qualitative comparison between InstructEngine and baseline methods (SDXL, ReFL, DPO, HPSv2) across six diverse text prompts.

