# OpenReview forum: "InstructEngine: Instruction-driven Text-to-Image Alignment"
_ICLR.cc/2026/Conference — ICLR 2026 Conference Withdrawn Submission_

### Official Review · Reviewer_jT2D · 2025-10-24

**Soundness:** 2
**Presentation:** 3
**Contribution:** 2
**Rating:** 4
**Confidence:** 3

**Summary:**

The paper introduces InstructEngine, a paradigm for aligning text‑to‑image diffusion models using fine‑grained, interpretable, instruction pairs rather than solely image‑level preference signals or opaque reward models. On DrawBench, InstructEngine improves the average score of SD v1.5 by ≈10.53% and SDXL by ≈5.30%, and attains >50% win‑rate in a 200‑image, 10‑annotator human study versus strong baselines. Ablations suggest each component (paired instructions, cross‑validation loss, image filtering) contributes materially; keeping discarded triples hurts performance.

**Strengths:**

* Encoding preferences at the instruction level (consistency, realism, aesthetics) makes alignment goals explicit and human‑readable—addressing opacity of single‑score reward models.
* The pipeline reduces human annotation cost by leveraging LLMs (taxonomy expansion, entity vetting, filtering) and shows favorable data‑efficiency curves compared to Pick‑a‑Pic and DiffusionDB selections at equal data budgets.
* The cross‑validation DPO formulation (instruction fixed, images contrasted) is well‑motivated and empirically supported (loss curves and ablations). [516_Instru...nstruction | PDF]

**Weaknesses:**

See Questions

**Questions:**

From Figure 4, I can see that the data construction method itself brings significant improvements compared to other datasets, which demonstrates the value of the proposed data pipeline. However, in Figure 7, for example, the “w/o Cross-val” variant achieves an ImageReward score of 0.7277. Comparing this to Table 1, this score is only slightly higher than the “Origin” baseline, even though “w/o Cross-val” already uses the new data (just without the cross-validation alignment). This seems somewhat inconsistent with the results in Figure 4, where the new data alone appears to provide a much larger gain. Could the authors clarify this apparent discrepancy? Is it due to differences in evaluation settings, or does it suggest that the cross-validation alignment is essential for fully leveraging the new data?

Inconsistencies number: Entities: “10,000” (Fig. 3) vs “15,000” (Sec. 3.2).

Typo in Eq? In Eq. (8), the second term is written as (Δₓw₂ − Δₓl₁) after defining Δₓl₂—likely a typo (should be Δₓl₂).

---

### Official Review · Reviewer_M63T · 2025-10-27

**Soundness:** 3
**Presentation:** 4
**Contribution:** 3
**Rating:** 4
**Confidence:** 3

**Summary:**

This paper introduces InstructEngine, a novel instruction-driven framework for aligning text-to-image generative models with human preferences. The method addresses key limitations in existing alignment approaches, such as reliance on expensive human-annotated preference data, the opacity of reward models, and the underutilization of textual information. Extensive evaluations demonstrate the benefits of InstructEngine.

**Strengths:**

1. Originality: The core idea of encoding preference information directly into contrasting text instructions is highly original, which offers a more interpretable and text-centric pathway for alignment.
2. Quality: The experimental design is rigorous, including comparative experiments, ablation studies, and human evaluations. The data construction process is highly automated, with well-defined and detailed categorization.
3. Clarity: The paper is well-structured and clearly written, with helpful visualizations of the taxonomy and data construction pipeline. The experimental section is also clear.
4. Significance: The work addresses the scalability and interpretability of text-to-image alignment. It greatly reduces the reliance on manual annotation and reward models, and has broad potential in t2i generative model tuning.

**Weaknesses:**

1. The preference dataset is generated using a single foundation model SDXL. This might introduce a potential source of bias, as the aligned models are effectively learning preferences distilled from SDXL's own generative characteristics.
2. The constructed preference pairs seem to exhibit obvious contrasts, as visualized in the paper. While this design ensures clear preference signals, it may oversimplify the learning task. This could lead to a ceiling effect in learning and reduce the model's robustness when faced with finer-grained distinctions.
3. The analysis and theoretical support for the cross-validation preference alignment algorithm are insufficient. A more rigorous theoretical explanation or additional experiments would strengthen this critical design choice.

**Questions:**

1. Since all training data are generated by SDXL, to what extent might the aligned models inherit SDXL’s inherent biases and limitations? Have you explored using multiple t2i models to increase dataset diversity and robustness?
2. The preference pairs appear to represent relatively “easy” contrasts with clear winners and losers. Could this design introduce a ceiling effect or reduce robustness when handling finer-grained distinctions?
3. How does the training cost of InstructEngine compare with existing methods?

---

### Official Review · Reviewer_5gtY · 2025-10-27

**Soundness:** 3
**Presentation:** 2
**Contribution:** 3
**Rating:** 6
**Confidence:** 4

**Summary:**

This work proposes an end-to-end instruction-driven alignment framework designed to better align text-to-image models with human preferences, without relying on costly large-scale manual preference scoring or additional reward models. Its primary contributions comprise automated preference data construction and Cross-Validation Alignment.

**Strengths:**

1. It introduces a novel paradigm that encodes preference information directly within fine-grained textual instructions, rather than relying solely on image-based feedback or external reward models.
2. It proposes an automated and scalable data construction pipeline, minimizing manual annotation costs while maintaining data diversity and quality.
3. It demonstrates adaptability to auto-regressive models such as Emu3.

**Weaknesses:**

1. The pipeline still relies on GPT-4o and BLIP for filtering and evaluation, which may introduce model bias and reinforce the underlying tendencies of these large models.
2. The pipeline discards text-based preference triples (two instructions and one image) due to empirical underperformance but lacks a formal theoretical explanation.
3. Some important human-alignment factors, such as safety, are not explicitly modeled. However, these factors are adopted in human evaluation. The motivation should be illustrated in detail.
4. Since SDXL serves as both a data generator and a target model, its inherent stylistic or semantic biases may propagate through the dataset, limiting the objectivity of the alignment and potentially capping performance gains. What about other models?

**Questions:**

Please refer to the weaknesses.

---

### Official Review · Reviewer_FLE4 · 2025-10-31

**Soundness:** 2
**Presentation:** 3
**Contribution:** 2
**Rating:** 2
**Confidence:** 5

**Summary:**

This work proposes InstructEngine which enables a different diffusion preference optimization framework wherein instead of just a winning and losing image pair for a given text prompt, this work constructs an opposite prompt which helps construct the losing pair. In addition, these two triplets are then used in standard diffusion loss to obtain fine-grained preference learning. For preference dataset constriction, it creates a taxonomy for image-to-text instructions by dividing T2I scenarios into 33 major themes with 20+ subtopics in each. The instruction prompts are generated from these to ensure diversity. Further, coarse-grained instructions along with opposite details of three dimensions are added for generating images with an off-the-shelf T2I model. This results in a preference dataset with ~25K samples.   Numerical evaluations on SD1.5 and SDXL showcase the efficacy of the proposed method and the proposed preference dataset.

**Strengths:**

- Dataset construction process looks high quality (except the choice of generation model).
- Alignment loss seems useful given that it would allow more fine-grained preference control due to the choice of available winning / losing pairs based on correct and opposite instruction prompts.

**Weaknesses:**

- Paper misses out on many strong baselines such as supervised fine-tuning (SFT) on the constructed dataset.
- Lack of evaluation on diverse T2I benchmarks such as DPG-Bench, GenEval, T2I-Comp, etc.
- Lack of comparison w.r.t. popular preference alignment methods such as MAPO, SPO, RankDPO, Flow-GRPO, etc. (see missing references)

**Questions:**

- How do you evaluate the consistency of the instructions against the generated images?
- Why not evaluate other benchmarks such as DPG-Bench, GenEval, T2I-Comp etc.?
- How does the proposed method help flow-matching based diffusion models such as SD3, Flux, Qwen-Image, etc. ?
- In Tab.1, how does the performance of the base model improve with simple supervised fine-tuning (SFT) on the full constructed dataset? Since it’s supposed to be significantly better in image-text alignment due to extensively filtered dataset construction and evaluation using GPT-4o.
- Why not use better models for image generation than SDXL? It seems quite old generation compared to the new generation models like Flux, Qwen-Image, etc.
- Human evaluation in Sec. 4.4 only uses 200 images. This evaluation set looks a bit too small. Have you done the user study on any larger set such as 1000+ images? Current sample set looks a bit too small to give meaningful comparison and conclusion on the efficacy of the proposed work.

Missing References:
- Margin-aware Preference Optimization for aligning diffusion models without reference (MAPO): https://arxiv.org/abs/2406.06424
- SPO : Step-aware Preference Optimization: Aligning Preference with Denoising Performance at Each Step https://arxiv.org/pdf/2406.04314v1
- Flow-GRPO— Training Flow Matching Models via Online RL : https://arxiv.org/pdf/2505.05470
- RankDPO — Scalable Ranked Preference Optimization for Text-to-Image Generation:  https://arxiv.org/abs/2410.18013
- Latent Preference Optimization (LPO) — Diffusion Model as a Noise-Aware Latent Reward Model for Step-Level Preference Optimization   : https://arxiv.org/pdf/2502.01051
- DSPO — Direct Score Preference Optimization for Diffusion Model Alignment https://openreview.net/forum?id=xyfb9HHvMe
- Bridging SFT and DPO for Diffusion Model Alignment with Self-Sampling Preference Optimization https://arxiv.org/abs/2410.05255

---

### Note · Authors · 2025-11-24

I have read and agree with the venue's withdrawal policy on behalf of myself and my co-authors.